# Antioxidant and Anti-Inflammatory Properties of Bioavailable Protein Hydrolysates from Lupin-Derived Agri-Waste

**DOI:** 10.3390/biom11101458

**Published:** 2021-10-04

**Authors:** Sergio Montserrat-de la Paz, Alvaro Villanueva, Justo Pedroche, Francisco Millan, Maria E. Martin, Maria C. Millan-Linares

**Affiliations:** 1Department of Medical Biochemistry, Molecular Biology, and Immunology, School of Medicine, Universidad de Sevilla, Av. Sanchez Pizjuan s/n, 41009 Seville, Spain; delapaz@us.es; 2Plant Protein Group, Food and Health Department, Instituto de la Grasa, CSIC. Ctra. de Utrera Km. 1, 41013 Seville, Spain; alvarovillanueva@ig.csic.es (A.V.); jjavier@cica.es (J.P.); fmillanr@ig.csic.es (F.M.); mcmillan@ig.csic.es (M.C.M.-L.); 3Department of Cell Biology, Faculty of Biology, Universidad de Sevilla, Av. Reina Mercedes s/n, 41012 Seville, Spain; 4Cell Biology Unit, Instituto de la Grasa, CSIC. Ctra. de Utrera Km. 1, 41013 Seville, Spain

**Keywords:** agri-lupin waste, by-products, bioeconomy, plant proteins, anti-inflammatory, antioxidant, bioavailability

## Abstract

Agri-food industries generate several by-products, including protein-rich materials currently treated as waste. Lupine species could be a sustainable alternative source of protein compared to other crops such as soybean or chickpea. Protein hydrolysates contain bioactive peptides that may act positively in disease prevention or treatment. Inflammatory responses and oxidative stress underlie many chronic pathologies and natural treatment approaches have gained attention as an alternative to synthetic pharmaceuticals. Recent studies have shown that lupin protein hydrolysates (LPHs) could be an important source of biopeptides, especially since they demonstrate anti-inflammatory properties. However, due to their possible degradation by digestive and brush-border enzymes, it is not clear whether these peptides can resist intestinal absorption and reach the bloodstream, where they may exert their biological effects. In this work, the in vitro cellular uptake/transport and the anti-inflammatory and antioxidant properties of LPH were investigated in a co-culture system with intestinal epithelial Caco-2 cells and THP-1-derived macrophages. The results indicate that the LPH crosses the human intestinal Caco-2 monolayer and exerts anti-inflammatory activity in macrophages located in the basement area by decreasing mRNA levels and the production of pro-inflammatory cytokines. A remarkable reduction in nitric oxide and ROS in the cell-based system by peptides from LPH was also demonstrated. Our preliminary results point to underexplored protein hydrolysates from food production industries as a novel, natural source of high-value-added biopeptides.

## 1. Introduction

The United Nations (UN) Sustainable Development Goals (SDGs) aim to shrink the global food waste generated per capita along the food supply chain by up to 50% by 2030 [1]. Currently, protein-rich agri-industrial waste is generated, among which is the flour or defatted flours from legumes and oilseed, and which constitute an important source for obtaining protein concentrate and isolates [2,3]. Green industries, such as those that produce industrial agricultural lupin, offer opportunities for more long-term benefits through sustainable regenerative farming practices. The industrial lupin market in Europe, which reached a size of 284 million in 2019, is expected to become profitable by 2027, growing at a rate of 4.39% in the forecast period 2020–2027 [4]. Growing concerns about food issues, as well as strict national and European regulations, make functional food an added value for the agri-food sector, given the economic impact of the commercialization of this type of product. In the literature, several works refer to the use of different parts of lupine such as whole grains and defatted flour, among others [5,6,7,8]. Lupin seed protein content is high (up to 44%) and of good quality, offering potential health benefits, and contributing to the sustainability of cropping systems [9,10,11].

Chronic inflammation and oxidative stress play a key role in the development and progression of many chronic diseases such as autoimmune diseases, metabolic disorders, cardiovascular diseases, central nervous system-related disorders, fibrosis, diabetes, obesity, and cancer [12,13,14]. Diet is one of the major risk factors for the development of chronic diseases, so the modification of diet could prevent or delay the onset of these diseases. In this sense, functional foods and nutraceuticals have emerged as potential tools to improve health and well-being, and reduce the risk or delay the onset of major diseases, in concomitance with a reduction in side effects inherent to synthetic pharmaceuticals. One of the components of functional foods are bioactive peptides, which mostly contain 3–20 amino acid units, and are often encrypted in the native sequence of proteins and can be released by digestive enzymes during food processing or by in vitro hydrolysis by proteolytic enzymes [15]. Bioactive peptides can be classified according to their mode of action as antimicrobial, antithrombotic, antihypertensive, opioid, immunomodulatory, mineral-binding, and antioxidative [16]. These multifunctional activities can occur through different molecular and signaling mechanisms, passive paracellular transport, transporter-mediated active transport (via the PepT1), and transcytosis [17]. The intestinal absorption of carbohydrates such as glucose and lipids is well known; however, most of the mediators involved in the transport of peptides at the intestinal level have not yet been identified.

Bioactive peptides are widely available from protein hydrolysates in the plant kingdom, considering that most of the population’s protein needs are covered with plant-based foods. On the other hand, food manufacturers produce large amounts of protein-rich by-products or waste from plants, with the main plant-protein sources being oilseeds, cereals, and legumes [18,19]. Thus, nutritional properties of natural protein hydrolysates as agri-food processing compounds have increased in interest during recent years. Indeed, proper nutrition may connect with the desirable concern about environmental impact derived from industrial activities [19]. One important aspect of bioactive peptides is their bioavailability. To exert their effects, peptides need to resist degradation by gastrointestinal proteases and brush-border peptidases, be absorbed through the intestinal epithelium, and reach the bloodstream in an active form [20,21]. Caco-2 is a cell line used to predict absorption in the small intestine, since differentiated Caco-2 cells maintain the morphology and function of mature enterocytes, and express brush-border peptidases and transporters that may affect the bioactivity of peptides [20,22].

In previous studies, we have described that lupin protein hydrolysates (LPHs) have potential anti-inflammatory properties due to the inhibition of enzymes involved in the inflammatory pathway [23] and the decrease in production and expression of pro-inflammatory cytokines in THP-1-derived macrophages [24] and osteoclasts [25]. The main aim of this manuscript was to study whether an LPH with anti-inflammatory properties may pass through the intestinal lumen and reach the bloodstream, so that bioactive peptides found in LPH are also bioavailable. Additionally, we studied the potential antioxidant effects of LPH peptides, measuring nitrite production and ROS levels. For these purposes, we tested an LPH in a co-culture system with Caco-2 cells and THP-1-derived macrophages stimulated with lipopolysaccharides (LPS). To the best of our knowledge, this is the first paper simultaneously measuring the bioavailability and bioactivity of plant-protein hydrolysates, an underexplored agri-food waste with health-promoting properties, in a cell-based system, providing opportunities for future research in tissue and organ models.

## 2. Materials and Methods

### 2.1. Materials and Reagents

Lupin protein isolate (LPI) was obtained in a pilot plant of vegetable proteins obtained from defatted lupin flour, as described by Millan-Linares et al. [23]. Alcalase® 2.4 L (2.4 AU/g), a non-specific serine endopeptidase, was provided by Novozymes (Bagsværd, Denmark). The cell types used were THP-1 monocytes (ATCC Number TIB-202) and Caco-2 (ATCC Number HBT-37). Culture media, fetal bovine serum (FBS), penicillin (P), and streptomycin (S) were from Gibco (Life Technologies SA, Spain). Phorbol 12-myristate 13-acetate (PMA) and LPS from *Escherichia coli* O55:B5 were purchased by Sigma Chemical Co. (St. Louis, MO, USA). Co-cultures were prepared in 12-well Millicell® plates with hanging cell-culture inserts 0.4 µm PET (Millipore Corporation, Billerica, MA, USA). The iScript cDNA Synthesis Kit was obtained from Bio-Rad Laboratories (Hercules, CA, USA). Brilliant II SYBR Green QPCR Master Mix was purchased from Agilent Technologies (Santa Clara, CA, USA). All other reagents were of analytical grade.

### 2.2. Preparation of Lupin Protein Hydrolysate

Hydrolysis was performed in a bioreactor of the pilot plant of vegetable proteins, under stirring at a controlled pH and temperature. LPI was suspended in distilled water (10% w/v) and hydrolyzed with Alcalase® for 15 min at pH 8, 50 °C, and E/S = 0.3 AU/g protein. The enzyme was inactivated through heating at 85 °C for 15 min. The supernatant obtained after centrifugation at 8000 rpm for 15 min constituted the LPH.

### 2.3. Analytical Methods

The protein concentrations were determined by elemental microanalysis as a percentage of nitrogen content x 6.25 using a Leco CHNS-932 analyzer (St. Joseph, MI, USA). Ash content was determined by the direct ignition method (550 °C for 25 h). Total dietary fiber was determined according to a method presented by Lee et al. [26]. Oil content was measured using the AOAC (Association of Official Analytical Chemists) method 945.16 [27]. Polyphenols and soluble sugars were measured using chlorogenic [28] and glucose [29] standard curves, respectively. Amino acid composition was evaluated according to a method proposed by Alaiz et al. [30]. Tryptophan content was analyzed as described by Yust et al. [31].

### 2.4. Caco-2 Culture

Caco-2 cells were cultured in 12-well cell culture inserts in Dulbecco’s Modified Eagle Medium, supplemented with 10% heat-inactivated FBS and 1% P/S. Cells were incubated at 37 °C under a modified atmosphere of 5% CO_2_ and given a fresh medium every 2–3 days. Cell monolayer integrity was monitored by trans-epithelial electrical resistance using a Millicell® ERS-2 voltammeter (Millipore). Inserts were used for 3 weeks after seeding, and had a resistance of at least 500 Ω/cm^2^.

### 2.5. THP-1 Culture

The human monocytic THP-1 cell line was cultured in a suspension of RPMI 1640 supplemented with 10% heat-inactivated FBS and 1% P/S. Cells were seeded at 3 × 10^5^ cells/well in 12-well plates and differentiated to macrophage-like cells by treatment for 4 days with PMA at 100 nmol/L [24].

### 2.6. In Vitro Availability Using Caco-2/THP-1 Co-Cultures

Inserts were transferred to 12-well plates containing THP-1-derived macrophages. Treatments, after in vitro stimulation with or without LPS (100 ng/mL), were carried out under the same standard incubation conditions, and were initiated by replacing the medium with fresh medium containing LPH at 0.1 and 0.5 mg/mL in the apical chamber. After 24 h, the supernatant in the basolateral side was recovered and RNA from THP-1-derived macrophages cells stimulated with LPS was extracted.

### 2.7. Reactive Oxygen Species Determination

For the determination of the intracellular concentration of ROS, the fluorescent probe, 2′,7′-dichlorofluorescein diacetate (DCFDA), a fluorogenic dye that measures hydroxyl, peroxyl, and other ROS activity within the cell, was used. After diffusion into the cell, DCFDA was deacetylated by cellular esterases into a non-fluorescent compound, which was later oxidized by ROS into 2′,7′-dichlorofluorescein (DCF), a highly fluorescent compound, quantified by flow cytometry at 485 nm (excitation) and 535 nm (emission) [32].

### 2.8. Nitric Oxide Assay

THP-1-derived macrophages were seeded in a 96-well plate at a density of 5 × 10^5^ cells per well and incubated overnight. Cells were pre-treated with both concentrations of LPH (0.5 to 10 mg/mL) for 24 h. After the treatments, culture supernatants were collected, and the nitric oxide (NO) concentration was measured using a Griess reagent assay kit (R&D Systems, Inc. Minneapolis, MN, USA), which measures the level of accumulated nitrite, a NO metabolite. Absorbance was determined to be 540 nm [32].

### 2.9. RNA Extraction and Analysis by RT-qPCR

Total RNA was extracted from LPS-stimulated-THP-1 cells using NucleoSpin RNA II. RNA quality was assessed using the OD_260_:OD_280_ ratio determined by a NanoDrop ND-1000 Spectrophotometer (Thermo Scientific, Waltham, MA, USA). One microgram of total RNA was subjected to RT-PCR testing to obtain cDNA according to the manufacturer’s protocol. The mRNA levels for specific genes were determined using an Mx3000P Real-Time PCR System (Stratagene, La Jolla, CA, USA). For each qPCR, 10 ng of cDNA template was added to the Brilliant SYBR Green qPCR Master Mix containing primer pairs for TNF-α, IL-6, IL-1β, and IL-10. The reference genes HPRT and GAPDH were used to correct for RNA concentration differences between the samples. The sequence of and information concerning the primers that were used in this study are as follows: TNF-α (NM_000594.3): 5′-TCCTTCAGACACCCTCAACC-3′ and 5′-AGGCCCCAGTTTGAATTCTT-3′ (reverse); IL-1β (NM_138712): 5′-GCTGTGCAGGAGATCACAGA-3′ and 5′-GGGCTCCATAAAGTCACCAA-3′; IL-6 (NM_001001928): 5′-GTTTGAGGGGGTAACAGCAA-3′ and 5′-GCTAACTGCAGAGGGTGAGG-3′; IL-10 (NM_000572.2): 5′-GTTCTTTGGGGAGCCAACAG-3′ and 5′-GCTCCCTGGTTTCTCTTCCT-3′; HPRT (NM_000194.2): 5′-ACCCCACGAAGTGTTGGATA-3′ and 5′-AAGCAGATGGCCACAGAACT-3′; and GAPDH (NM_002046.4): 5′-GAGTCAACGGATTTGGTCGT-3′ and 5′-TTGATTTTGGAGGGATCTCG-3′. All amplification reactions were performed in triplicate. The magnitude of the change in mRNA expression for the candidate genes was calculated using the standard 2^–(ΔΔCt)^ method. All data were normalized to the endogenous reference gene (HPRT and GAPDH) level, and expressed as a relative value of the control. The fold change was determined as follows: individual fold changes for all pairs of samples derived from the two groups to be compared were calculated, and the median of these individual fold changes represents the overall fold change for the given gene.

### 2.10. Enzyme-Linked Immunosorbent Assay (ELISA)

TNF-α, IL-1β, IL-6, and IL-10 concentrations in cell culture supernatants were quantified by commercial ELISA kits, according to the manufacturer’s instructions.

### 2.11. Statistical Analysis

All values in the figures and text are expressed as the arithmetic mean ± SD. All experiments were carried out in triplicate, and repeated at least three times independently, with the exception of obtaining protein isolate and hydrolysates. The statistical analysis was performed with Graph Pad Prism Version 6.01 software (San Diego, CA, USA). The data were analyzed using a one-way analysis of variance (ANOVA), followed by Bonferroni tests for multiple comparisons. Differences were considered significant at *p* < 0.05.

## 3. Results and Discussion

### 3.1. Chemical Characterization of LPH

In previous studies, we reported that the hydrolysis of LPI with Alcalase® led to the production of LPHs with potential anti-inflammatory properties due to the inhibition of enzymes involved in the inflammatory route [23]. In this previous work, several hydrolysis timeframes were tested, and we concluded that LPH produced after 15 min showed the best effect. Specifically, this LPH inhibited 90% COX-2, 22% thrombin, and 50% transglutaminase activity, whereas LPI inhibited 65% COX-2, 5% thrombin, and inhibition of transglutaminase activity was not detected [23]. Moreover, this LPH induced the downregulation of TNF, IL-1β, and IL-6 in THP-1-derived macrophages [24].

The chemical composition of LPI and LPH is shown in Table 1. The protein content of LPI is above 86% on a dry basis; this value is similar to that obtained by other authors [33,34], and much higher than that of the starting flour with a protein richness of 39.91%. The chemical composition of LPH was similar to LPI; the main difference between the two samples was ash content, which was higher in LPH. This is a normal consequence of the addition of alkali during the hydrolytic process to maintain the pH constant [35]. Consequently, the other components of LPH, with the exception of polyphenols, decreased. The amino acid composition of both products did not show significant differences (Table 2); this evidences that enzymatic hydrolysis does not alter the nutritional value of original proteins [35]. Furthermore, LPI and LPH showed an amino acid profile typical of lupin proteins, being the main components glutamic acid/glutamine, arginine, and aspartic acid/asparagine, and the minor ones sulfur amino acids and tryptophan [36]. Both products meet the nutritional requirements proposed by FAO/WHO, with the exception of sulfur amino acids, which are common in legume proteins [35,37]. Considering that LPI and LPH showed similar compositions, the increase in anti-inflammatory activity in LPH should be ascribed to the release, due to the hydrolytic process, of specific peptides encrypted in the protein sequences.

### 3.2. LPH Reduces Production of ROS and Nitrites in LPS-Stimulated THP-1-Derived Macrophages

Caco-2 monolayers are widely used to predict transport via different pathways across the intestinal lumen [21,39,40]. In order to study the bioavailability of antioxidant and anti-inflammatory peptides in LPH, co-cultures of Caco-2 and THP-1-derived macrophage in 12-well Millicell® plates with hanging inserts were used. This system has two compartments, and a Caco-2 monolayer separates the apical zone, corresponding to the intestinal lumen, from the basolateral one, corresponding to the intestinal vascular and blood circulation system (Figure 1).

The co-cultures were treated with 0.1 and 0.5 mg/mL of LPH and, after 24 h, the basolateral supernatant was recovered to measure biological activities. Two controls without LPH were studied, and consisted in THP-1-derived macrophages without LPH (control) and with LPS stimulation (control+LPS). As shown in Figure 2A, intracellular ROS production decreased by less than half in LPS-induced conditions in the presence of both LPH doses, almost restoring the control percentage after treatment with 0.5 mg/mL of LPH. In line with these effects, Figure 2B shows an LPH dose-dependent decrease in nitrite release to the culture medium in LPS-stimulated cells, under 50% nitrite values in the presence of 0.1 mg/mL LPH, and a similar percentage to non-stimulated conditions after 0.5 mg/mL LPH treatment. As already stated, oxidative stress and inflammatory microenvironments are related to chronic diseases [14]. Antioxidant activities help to restore balance conditions caused by inflammation, cancer, or aging. Several protein hydrolysates and peptides with antioxidant effects from animals and plants, such as milk, fish, sesame or wheat brans, hemp or rapeseed, corn, or soy, have been tested both in vitro and in vivo with promising results [18,19,41]. Cell-based systems have also been useful to elucidate the detailed antioxidant mechanism of milk peptides [42,43], fish or seaweed protein hydrolysates [44], and hemp seed or rice endosperm proteins. Nevertheless, only a few studies, including those involving rapeseed hydrolysates, have been assessed in animal models for verifying antioxidant activities in biopeptides [14].

Our data focus on the antioxidant potential of LPH as a natural source of biopeptides that may exert their activity at target sites after crossing the gastrointestinal barrier. Bioavailability and oxidative balance recovery, as shown by LPH, provide an opportunity for the development of functional foods and the reduction in wastes from agri-industrial activities.

### 3.3. Effect of LPH on mRNA Expression of Pro-Inflammatory Cytokines

In order to investigate whether LPH peptides could be absorbed through intestinal lumen and decrease cytokine production by the regulation of gene expression, we measured mRNA expression of TNF-α, IL-1β, IL-6 and IL-10 in THP-1-derived macrophages co-cultured with Caco-2. Pro-inflammatory cytokine (TNF-α, IL-1β, and IL-6) mRNA expression increased threefold when THP-1 cells were stimulated with LPS. The addition of LPH to the culture medium decreased its mRNA expression up to the control values; in other words, the treatment with LPH returned to the inflammatory conditions in the unstimulated THP-1-derived macrophages. The effect of LPH was even more drastic in IL-1β mRNA expression, since the values were lower than in the control group (Figure 3). There were no significant differences between the two LPH concentrations tested. On the other hand, LPH increased mRNA levels of anti-inflammatory IL-10 cytokine after 24 h of treatments, but only at a concentration of 0.5 mg/mL, indicating that the anti-inflammatory effect is dependent on the concentration of the hydrolysate.

The inflammatory responses that underlie many chronic diseases involve mediators such as pro-inflammatory cytokines, chemokines, and reactive oxygen species, affecting a wide range of cells. Due to the complex and multisystem events that occur during inflammation, it is necessary to develop many approaches to investigate both the expression and the presence of these markers in cell-based systems [14]. Protein-rich by-products from animal- and plant-processing industries have previously demonstrated a key role during inflammation due to the presence of bioactive isolated peptides or hydrolysates. With the exception of milk and egg components, in general, more studies with plant material have been carried out [14]. Our results of LPH decreasing the expression of pro-inflammatory cytokines are similar to others shown in plant-protein hydrolysates, such as wheat, gluten, or hemp [45,46].

### 3.4. Effect of LPH on the Production of Pro- and Anti-Inflammatory Cytokines

Anti-inflammatory activity was tested by measuring the secretion of pro-inflammatory cytokines TNF-α, IL-1β, and IL6, and anti-inflammatory cytokine IL-10 by THP-1-derived macrophages, since they have been described as the major cytokines involved in inflammation [47]. The concentration of pro-inflammatory cytokines in the supernatant of THP-1-derived macrophages stimulated with LPS (without LPH) was significantly raised in comparison with the control without LPS, demonstrating that THP-1-derived macrophages cells were effectively induced towards an inflammatory state. The treatment with LPH reduced the secretion of TNF-α, IL-1β, and IL-6 up to control levels. There were no statistic differences when varying amounts of LPH were used, since both of them decreased the concentration of TNF-α, IL-1β, and IL-6 by 70, 40, and 45%, respectively (Figure 4). These results are even higher than those previously reported by our research group, since we found that TNF-α production was reduced by 31% by treatment with the same LPH that was used in this study [24]. However, results cannot be directly compared, since THP-1 cells were differentiated to macrophages with PMA in Millan-Linares et al. [24]. LPH peptides crossed Caco-2 monolayers, and the modifications produced during this process, if any, did not negatively affect the bioactive properties. Similar conclusions have been described about the potential absorption of lupin peptides with hypocholesterolemic activity in human intestinal cells [21,48]. Furthermore, it has been reported as a possible selective absorption of the most bioactive components as a way to explain higher bioactivity of peptides transported by Caco-2 cells in the basolateral compartment than in the starting sample [21]. Regarding the anti-inflammatory cytokine IL-10, only LPH at the highest concentration showed a significant increase in its production in relation to LPS-stimulated THP-1-derived macrophages. It has been previously described that LPH has an important effect on the production and expression of pro-inflammatory cytokines, but not on anti-inflammatory cytokines [24]. The results of cytokine mRNA expression are consistent with cytokine production in the supernatant of culture medium, indicating that LPH acts at both the transcriptional and post-transcriptional levels. The same result was found when THP-1-derived macrophages were treated with a peptide purified from LPH [49].

## 4. Conclusions

We have demonstrated that antioxidant and anti-inflammatory peptides in LPH are potentially resistant to the gastrointestinal tract and may reach the bloodstream to exert their beneficial effects. Bioactivity of LPH from protein-rich by-products created during agri-industrial processes emphasize the valuation of traditional wastes as high-added-value products reducing the environmental impact of human activities. Due to the strong relationship between chronic inflammation, involving oxidative stress, and several chronic diseases, LPH (bioavailable and bioactive) may be excellent ingredients of functional foods or nutraceuticals, as a tool for fighting against these diseases and their risk factors, minimizing side effects compared to traditional therapies. Our study points to specific receptors and signaling pathways that mediate the anti-inflammatory and antioxidant molecular mechanisms of beneficial actions of lupin hydrolysates. However, further studies regarding peptides sequences, molecular receptors, and immunogenicity, must be performed to confirm these properties in vivo and translate this knowledge to a tissue and organ level, to fully exploit the potential of plant industry by-products which are rich in proteins and peptides.

## Figures and Tables

**Figure 1 biomolecules-11-01458-f001:**
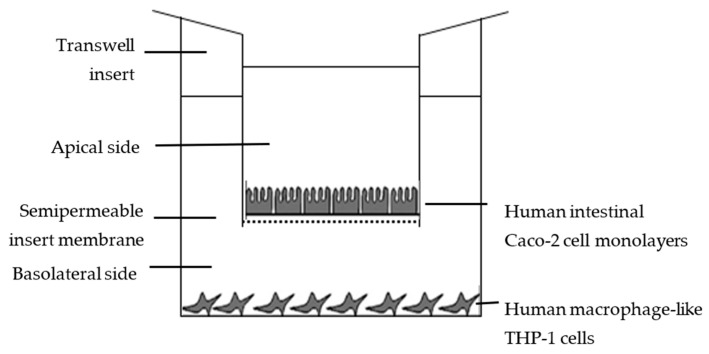
Schematic representation of the co-culture system of Caco-2 monolayer and LPS-stimulated THP-1-derived macrophages.

**Figure 2 biomolecules-11-01458-f002:**
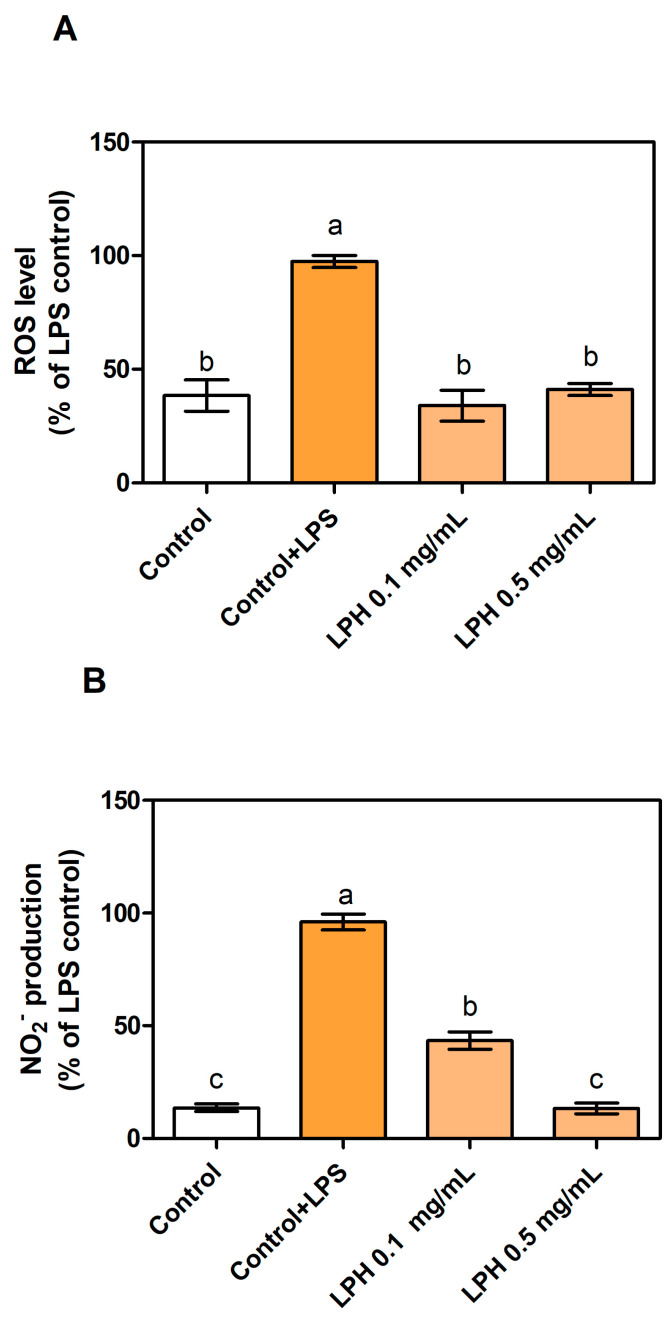
Intracellular ROS (**A**) and nitrite (**B**) production, expressed as percentage of fluorescence/absorbance after 24 h incubation with or without LPS (100 ng/mL) and LPH at 0.1 and 0.5 mg/mL. Values are presented as means ± SD (*n* = 3) and those marked with different letters are significantly different (*p* < 0.05).

**Figure 3 biomolecules-11-01458-f003:**
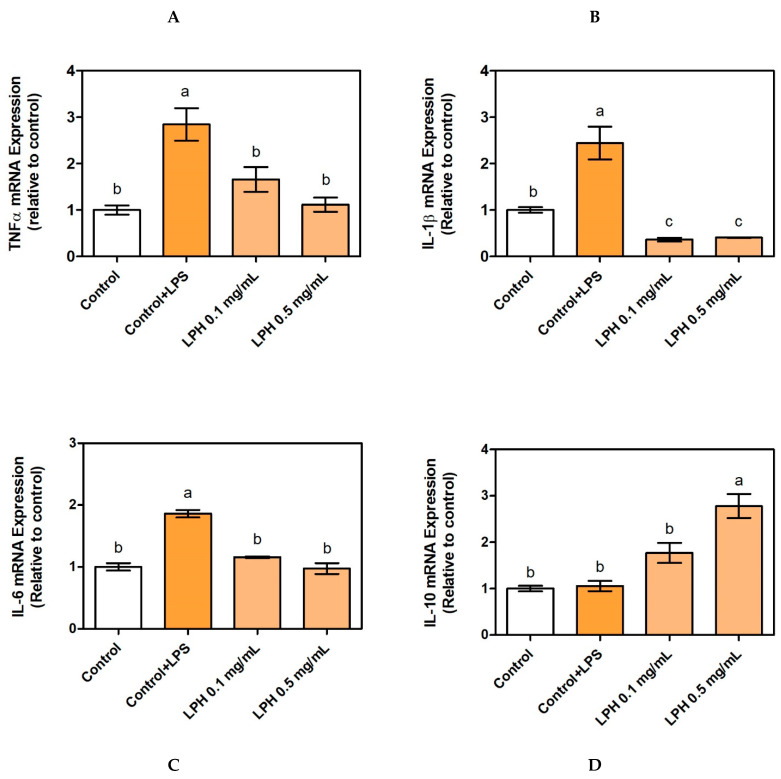
TNF-α (**A**), IL-1β (**B**), IL-6 (**C**), and IL-10 (**D**) mRNA expression in THP-1-derived macrophages co-cultured with Caco-2 cells in Millicell® system and treated with or without LPS (100 ng/mL) and LPH at 0.1 and 0.5 mg/mL. Values are presented as means ± SD (*n* = 3) and those marked with different letters are significantly different (*p* < 0.05).

**Figure 4 biomolecules-11-01458-f004:**
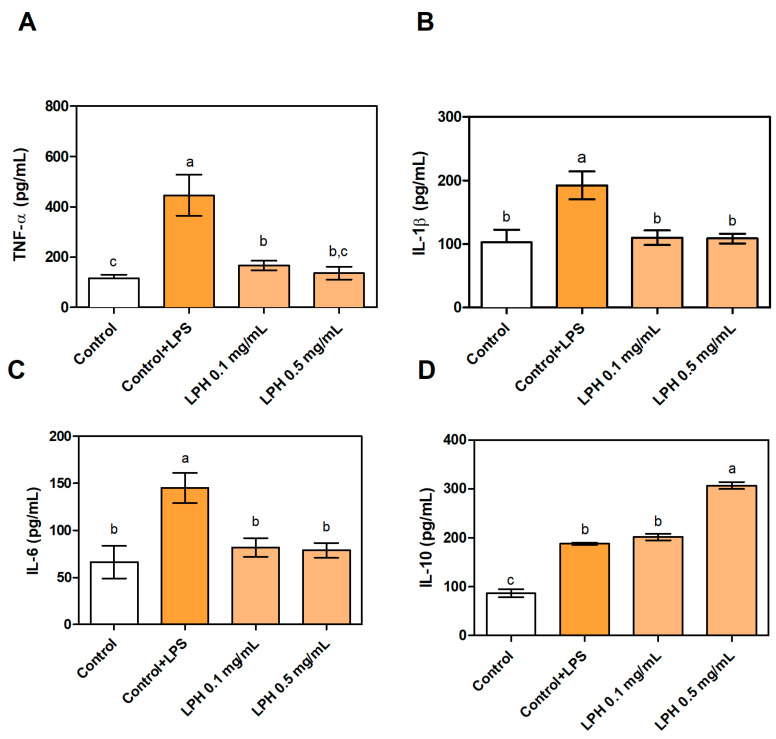
Production of (**A**) TNF-α, (**B**) IL-1β, (**C**) IL-6, and (**D**) IL-10 by THP-1-derived macrophages (Control), LPS-stimulated THP-1-derived macrophages (Control+LPS), and Control+LPS treated with 0.1 (LPH 0.1) and 0.5 mg/mL (LPH 0.5) of LPH. In all cases, THP-1 cells were co-cultured with Caco-2 cells in Millicell® systems. Values marked with different letters are significantly different (*p* < 0.05).

**Table 1 biomolecules-11-01458-t001:** Chemical composition of LPI and LPH. Data, expressed as percentage in dry basis, are mean ± standard deviation of three determinations.

(%)	LPI	LPH
Protein	86.72 ± 0.13	83.70 ± 0.09
Ash	0.78 ± 0.13	8.98 ± 0.09
Fibre	5.97 ± 0.34	0.97 ± 0.02
Oil	5.14 ± 0.17	1.15 ± 0.01
Soluble sugars	0.04 ± 0.00	0.02 ± 0.00
Polyphenols	0.01 ± 0.00	0.06 ± 0.00
Others ^1^	1.28	5.12

^1^ measured as 100-protein-ash-fibre-oil-soluble sugars-polyphenols.

**Table 2 biomolecules-11-01458-t002:** Amino acid composition of LPI and LPH. Data, expressed as percentage of amino acids on total amino acid content, are mean ± standard deviation of three determinations.

	LPI	LPH	FAO/WHO [38]
Asp+ Asn	10.68 ± 0.27	10.27 ± 0.12	
Glu+Gln	23.06 ± 0.40	24.47 ± 0.17	
Ser	5.96 ± 0.05	5.97 ± 0.16	
His	2.39 ± 0.15	2.36 ± 0.01	1.5
Gly	4.47 ± 0.17	4.52 ± 0.06	
Thr	3.88 ± 0.19	4.05 ± 0.02	2.3
Arg	11.78 ± 0.02	11.60 ± 0.06	
Ala	3.80 ± 0.04	3.89 ± 0.07	
Pro	0.75 ± 0.01	0.75 ± 0.01	
Tyr	4.27 ± 0.39	4.42 ± 0.12	
Val	3.98 ± 0.47	3.47 ± 0.07	3.9
Met	0.37 ± 0.00	0.44 ± 0.15	2.2 ^1^
Cys	0.78 ± 0.13	0.56 ± 0.19	
Ile	4.83 ± 0.04	4.45 ± 0.01	3.0
Leu	8.71 ± 0.03	8.55 ± 0.05	5.9
Phe	5.06 ± 0.01	4.95 ± 0.01	3.8 ^2^
Lys	4.87 ± 0.00	4.92 ± 0.02	4.5
Trp	0.38 ± 0.03	0.35 ± 0.02	

^1^ Met + Cys, ^2^ Phe + Tyr.

## Data Availability

Not applicable.

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
