# Peer review of "Antioxidant and Anti-Inflammatory Properties of Bioavailable Protein Hydrolysates from Lupin-Derived Agri-Waste"

_biomolecules, 2021, doi:10.3390/biom11101458_

Round 1

Reviewer 1 Report

Overall, the article is well organized and its presentation is good. However, some minor issues need to be improved:

  • Line 128 - 5% CO2? Please add information
  • Lines 252 - please use italics for in vitro and in vivo
  •  References - please use italics for latin names and unify references style.
  • In the introduction section, it is suggested to refer to works on the use of different parts of lupine (whole grains, deffated flour, etc.) and technological treatments (e.g. fermentation) and their antioxidant activity:
  • Bartkiene, E.; Sakiene, V.; Bartkevics, V.; Juodeikiene, G.; Lele, V.; Wiacek, C.; Braun, P.G. Modulation of the nutritional value of lupine wholemeal and protein isolates using submerged and solid-state fermentation with Pediococcus pentosaceus strains. Int. J. Food Sci. Technol. 2018, 53, 1896–1905.
  • Łopusiewicz, Ł.; Drozłowska, E.; Trocer, P.; Kwiatkowski, P.; Bartkowiak, A.; Gefrom, A.; Sienkiewicz, M. The Effect of Fermentation with Kefir Grains on the Physicochemical and Antioxidant Properties of Beverages from Blue Lupin (Lupinus angustifolius L.) Seeds. Molecules 2020, 25, 5791
  • Karamać, M.; Orak, H.H.; Amarowicz, R.; Orak, A.; Piekoszewski, W. Phenolic contents and antioxidant capacities of wild and cultivated white lupin (Lupinus albus L.) seeds. Food Chem. 2018, 258, 1–7
  • Khan, M.K.; Karnpanit, W.; Nasar-Abbas, S.M.; Huma, Z.-E.; Jayasena, V. Phytochemical composition and bioactivities of lupin: A review. Int. J. Food Sci. Technol. 2015, 50, 2004–2012

Reviewer 2 Report

The manuscript “Antioxidant and anti-inflammatory properties of bioavailable protein hydrolysates from lupin-derived agri-waste” provides a good contribution for evaluating the antioxidant and anti-inflammatory properties of lupin protein hydrolysates. However, a few minor comments and suggestions of changes are included below.

Title

I would suggest this alternative title “Antioxidant and anti-inflammatory properties of protein hydrolysates from lupin-derived agri-waste”

Abstract

Line 30 – Are there "underexploited protein hydrolysates from food production industries"? Please clarify.

Introduction

Lines 42-49 – Please revise these sentences for clarification.

Line 61 - I think it would be better to talk about bioactive peptides that exhibit the different biological activities because many of these peptides are multifunctional as you know.

Line 63 – Which process? Please clarify.

Lines 72-75 – Please revise these sentences for improvement.

Line 76 – I suggest replacing “controversial” by “important”, for instance.

Lines 86-95 – In these sentences there is a mixture of objectives and methodologies followed in this study. Thus, a clear separation would be advisable.

Materials and Methods

Line 104 – I think that it is “were purchased”.

Line 120 – The reference to AOAC should be indicated.

Line 130 – It is voltohmmeter.

Line 135 – I suggest replacing “be treating” by “treatment”.

Results and Discussion

Line 199 – This is a minor comment but I suggest rephrasing this sentence because the times of hydrolysis were not tested but the products of course.

Line 233 – I think that it is “a Caco-2 monolayer”. Please check.

Figure 1 – An even smaller comment. The line in front of “Apical side” should be longer to really indicate the “apical side”.

Author Response

This manuscript is a resubmission of an earlier submission. The following is a list of the peer review reports and author responses from that submission.

Round 1

Reviewer 1 Report

The purpose of this work was to be investigated the bio-efficacy of some agri-waste derived lupin peptides under the light of their presumable bioavailability. In more detail, they tried to elucidate whether lupine protein hydrolysates could survive under the activity of the gastrointestinal proteases and brush-border peptidases and finally be absorbed through the intestinal epithelium, and reach the bloodstream, most important, in an active form which in this case expected to be antioxidant and anti-inflammatory. The chemical composition of lupin protein isolates and lupin protein hydrolysates was determined. Caco-2 monolayers were used to predict transport by different pathways across the intestinal lumen. Furthermore, concentrations of intracelular Reactive oxygen Species and nitric were used as a tool to evaluate the antioxidant activity of the bioavailable hydrolysates. The anti-inflammatory activity was studied via their effect on mRNA expression of pro-inflammatory cytokines. According to my opinion, the present work seems interesting, providing useful and promising data concerning a very important issue in the field of natural products meaning, the bioavailability of, the in vitro active, molecules either isolated from natural sources like plants or from their wastes. Overall, the experimentation seems solid while authors' conclusions compile to data extracted although I strongly agree that work has to be done in the future regarding peptides sequences, molecular receptors, and immunogenicity.

As a result, I believe that present work merit to be published. 

Reviewer 2 Report

Interesting article, however, the experimental design and statistical analysis are a bit doubtful.

  1. How many times did the authors repeat the Alcalase digestion? not clear from the manuscript if they do it in duplicate or triplicate then do they consider it as technical replicate?
  2. Once the digestion is complete they took the sample for cell culture study, in that they claim to have n=3 data. Now the question is n=3 for all the technical replicates of the digestion? Now if they treat each digestion as technical replicate and take them to the cell culture then do they have biological replicates in cell culture? If so then how they are presenting that data?
  3. Clearly, the design of experiments needs more details.
  4. In the figures, all those bar graphs please show the individual values along with the SD, one-side error bars are misleading.
  5. The LPS is not a good stimulator of ROS and the ROS level presented here is for only a one-time point? Again ROS generation is a dynamic process and certainly presenting a single time point could be biased.
  6. Figure 2 the x-axis level is different from all other figures- be consistent.
  7. In Figure 3 do they represent fold change data, it is advisable to clearly state the fold-change calculations in the methods section.

Reviewer 3 Report

In the manuscript titled “Evaluation of antioxidant and anti-inflammatory properties of agri-waste derived lupin bioavailable peptides”, Sergio Montserrat-de la Paz et al. have investigated the bioavailability of lupin-derived bioactive peptides as well as their antioxidant and anti-inflammatory activity. 

Major comments:

  1. The novelty of this study appears modest. Pulse-derived bioactive protein and peptides have been widely studied especially the antioxidant and anti-inflammatory activity. The present study focused on lupin, which is also a pulse product and there was no information highlighted the significance or difference of lupin and its-derived peptide compare with other pulse product. Thus, the overall novelty of the present study should be improved.
  2. I’m confused about the experimental material lupine protein hydrolysates (LPHs). The author mentioned that the LPHs were prepared by lupine protein isolate (LPI). Can LPI be considered an agri-waste? What is the main application of the lupine products currently? The authors should provide a brief introduction of the lupine and lupine industry in the Introduction to give a background and therefore provide a sound rationale.
  3. Why did the author choose Alcalase but not other enzymes to prepare the hydrolysate? The present study was trying to explore the bioavailability, I think pepsin and trypsin/pancreatin will be more reasonable.
  4. All experiments were developed using lupine protein hydrolysates, which were a mixture of various peptides. To study the potential bioactivities, the authors should identify the peptides, which can path through the intestinal lumen and exhibit the antioxidant and anti-inflammatory effect. Otherwise, the present study is quite similar to the published studies (reference [11][12][13]).

Minor comments:

  1. The authors used “lupin” in the title, but “lupine” in the text. Please keep consistent.
  2. In Figure 2, the concentration of LPH was 0.1 and 0.5 mg/mL, while in other figures, the unit changed to 100 and 500 μg/mL. Please keep consistent.

Reviewer 4 Report

This is a relevant topic in the field of upcycling of byproducts for value-added health applications. The cell culture work was well conducted, but the characterization part must be further developed to strengthen the study's contribution to the field.

Major issue

- Peptide characterization is missing in the study. Crude chemical (Table 1) and amino acid compositions (Table 2) are not relevant in the bioactivity reported. Authors need to identify all the peptides present in the hydrolysate and fraction that crossed the Caco-2 cells into the basolateral side. Identification of the peptides, especially the latter, would be the most significant finding of the study as it would facilitate target identification and nutraceutical development from the lupin(e) material.

Minor comments

- The title reads like a proposal instead of a research report. Please revise.

- Lupin and lupine were used interchangeably. Be consistent.

- Section 3 should be Results and Discussion, not Results.

- In vitro cellular (Caco-2 cell) uptake/transport was mischaracterized as bioavailability. The latter requires the use of whole organisms. Please correct.

- The claim on lines 73-76 is not relevant as several studies have been conducted for so many food-derived peptides, and the different source here is really not that important.

- Introduction should include what is already known about transepithelial transport of bioactive peptides.